# (Dynamic) Prompting might be all you need to repair Compressed LLMs

## Abstract

Large language models (LLMs), while transformative for NLP, come with significant computational demands, underlining the need for efficient, training-free compression. Notably, the reliability of perplexity as a benchmark for compressed model efficacy is in question, as our tests using LLaMA-7B and OPT-6.7b reveal a significant performance drop in several realistic downstream tasks, underscoring the disparity between perplexity as a performance indicator and real-world performance. Investigation into the trade-off between resource-intensive post-compression re-training highlights the prospect of prompt-driven recovery as a lightweight adaption tool. However, existing studies, confined mainly to perplexity evaluations and simple tasks, fail to offer unequivocal confidence in the scalability and generalizability of prompting. We tackle this uncertainty in two key ways. First, we uncover the vulnerability of naive prompts in LLM compression as an over-reliance on a singular prompt per input. In response, we propose *inference-time dynamic prompting* (IDP), a mechanism that autonomously chooses from a set of curated prompts based on the context of each individual input. Second, we delve into a scientific understanding of why *"prompting might be all you need post-LLM compression"*. Our findings suggest that compression does not irretrievably erase LLM model knowledge but displace it, necessitating a new inference path. IDP effectively redirects this path, enabling the model to tap into its inherent yet displaced knowledge and thereby recover performance. Empirical tests affirm the value of IDP, demonstrating an average performance improvement of 1.24% across nine varied tasks spanning multiple knowledge domains.

## 1 Introduction

Large language models (LLMs) have demonstrated exceptional proficiency in language generation and reasoning, rivaling human capabilities. The advent of models like GPT-4 (OpenAI, 2023) and tools such as ChatGPT signifies a major milestone along this trajectory, positioning themselves as pivotal assets in various industries. However, the escalating size of these models presents critical computational challenges, impeding efforts towards their widespread adoption (Chen et al., 2023).

In response to the growing requirements and the associated computational loads imposed by large language models (LLMs), techniques such as *quantization* and *sparsification* have garnered significant attention and resources. Quantization involves fine-tuning the bit-wise precision of a model to reduce its size, while sparsification entails eliminating redundant operations by nullifying weight or activation elements. Traditional methods like pruning and quantization typically necessitate a post-compression re-training step, be it iterative or one-shot, to restore performance (Han et al., 2015). Regrettably, given the scale of modern LLMs, even a one-shot re-training approach after compression is becoming prohibitively costly, underscoring the pressing demand for *training-free compression*. Recent endeavors, exemplified by works such as GPTQ (Frantar et al., 2022) and SparseGPT (Frantar & Alistarh, 2023), promise nearly unaltered accuracy, often assessed through the perplexity metric. However, our experiments, as depicted in Figure 2, reveal a noticeable drop in performance for LLaMA-7B (Touvron et al., 2023) and OPT-6.7b (Zhang et al., 2022) post-compression in several realistic downstream tasks. This underscores the disparity between perplexity as a performance indicator and real-world performance and strongly indicates a loss of knowledge within compressed LLMs. Thus, there is still a persisting need for performance recovery post-compression beyond parameter-tuning.

In an effort to reconcile the trade-off between the resource-intensive post-compression re-training and the observed real-world performance decline (despite nearly intact perplexity scores), a recent study by Xu et al. (2023) highlighted the potential of lightweight prompt adaptation to recover performance of compressed LLMs (either through direct prompt-tuning or by repurposing prompts tuned in alternative contexts). While the prospect of prompt-driven recovery is enticing, the study's assessment primarily relies on perplexity evaluations and relatively straightforward datasets. Upon subjecting their approach to complex downstream tasks and real-world metrics (as detailed in Section 3), we uncover similar performance caveats like the original GPTQ (Frantar et al., 2022) or SparseGPT (Frantar & Alistarh, 2023) methodologies. This performance-to-perplexity gap leaves

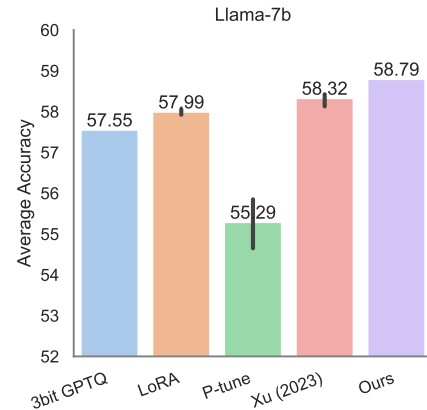

Figure 1: We map the average performance over nine tasks for various fine-tuning methods and ours on 3-bit quantized Llama-7b.

room for skepticism regarding the effectiveness of prompting for LLM post-compression performance recovery despite its appealing simplicity. Thus, in this paper, we ask: *Can prompting **scale** and **generalize** beyond rudimentary cases?*

## 1.1 OUR CONTRIBUTIONS

The first part of our work introduces innovative approaches aimed at providing a resounding affirmation to the pivotal question posed above:

- **Identifying Naive Prompt's Failure Modes in LLM Compression:** Our investigation uncovers a severe limitation in the performance of naive post-compression prompting (Xu et al., 2023). This limitation arises from an over-reliance on a singular prompt across diverse input formats and multifaceted knowledge domains, indicating a need for greater prompt diversity tailored to specific inputs.

- **Simple Fix by Dynamic Prompting:** To tackle the challenge of prompt diversity without introducing excessively long and resource-intensive prompts, we introduce *inference-time dynamic prompting* (**IDP**). This strategy empowers a compressed LLM to dynamically and autonomously select prompts from a curated set based on individual inputs. This dynamic adaptation allows inputs to incorporate relevant contextual cues without the need for manual intervention. Notably, besides parameters associated with the prompts themselves, IDP adds no additional parameters to prompting, and the per-instance compute cost remains nearly unchanged compared to using a fixed prompt for all inputs.

- **Strong Downstream Performance with Real-world Metrics and at Scale:** Our IDP approach demonstrates a substantial improvement in the performance of compressed LLMs on real-world tasks, compared to the original compressed models, the naively prompted model in Xu et al. (2023), as well as compressed models re-trained using LoRA, all while requiring no additional training parameters. Figure 1 highlights our competitiveness on 3-bit quantized Llama-7b.

The second part of our work delves into a scientific understanding of why *"prompting is all you need post-LLM compression"* and can perform on par with or even surpass re-training. This is an unexplored question to our best knowledge, and we present two hypotheses:

- *Null hypothesis (H0)*: Compression irreversibly impairs the model's knowledge, and prompting/IDP recovers this knowledge from scratch, through downstream data-driven learning, akin to re-training.

- *Alternative hypothesis (H1)*: Compression does not permanently erase inherent knowledge, but rather displaces it within the model, rendering the original inference path ineffective. Prompting/IDP enables the **"redirection"** of the inference path to reutilize this existing yet displaced knowledge in the compressed LLM, resulting in performance recovery. This rep-

resents a distinct form of recovery compared to re-training, which relies on **"re-learning"** the knowledge from data.

Our extensive and meticulously controlled experiments provide **strong support for (H1)** over (H0). For instance, utilizing layer-wise cosine similarity (refer to Figure 6), we observe that while prompt-tuning's attention patterns diverge from the baseline, the re-trained model exhibits substantially more similar attention patterns to the baseline, even with consistent outcomes among all three. Additionally, referring to Figure 7, we demonstrate that IDP's effectiveness remains robust even with low numbers of tokens per prompt. Collectively, these experiments endorse that prompts excel in redirecting token attention, tapping into pre-existing knowledge rather than introducing novel information to compressed LLMs.

## 2 MOTIVATION AND PRIOR WORKS

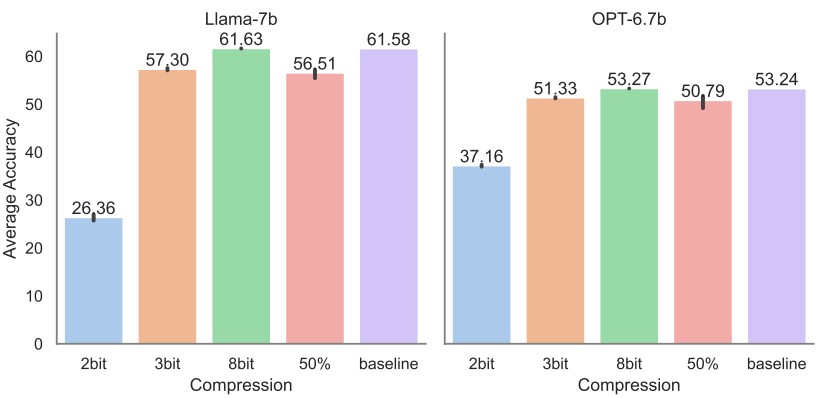

Figure 2: This figure presents a comparative analysis of the performance of compressed models using GPTQ for quantization and SparseGPT for pruning. The models were compressed leveraging either C4 or Wikitext datasets. Their average performance is depicted across a spectrum of nine tasks, each representing diverse knowledge domains.

**LLM inference bottleneck** LLMs such as those presented by Zhang et al. (2022); Touvron et al. (2023) predominantly utilize a decoder-only, autoregressive architecture, where token generation unfolds sequentially, and the creation of each subsequent token relies on the outputs generated before it. A significant portion of the computational time in LLMs is attributed to the decoding phase (Frantar et al., 2022; Liu et al., 2023). This phase is memory-constrained, typically operating with a single batch size. This memory-intensive nature is further corroborated by Liu et al., highlighting that generating a sequence is more time-consuming and than processing a sequence of equivalent length. This discrepancy arises from the I/O latency encountered during the loading of model parameters. While one might contemplate engineering in-memory solutions tailored for LLMs, as suggested by Sheng et al. (2023), such strategies do not tackle the underlying computational and memory challenges posed by these models.

**LLM compression solutions & caveats** Compression techniques directly address the challenges of size and latency inherent to LLMs by targeting the model's parameters. Broadly, these techniques are grouped into two main categories: compression-aware training and post-training compression. The latter, post-training compression, holds particular appeal for exceptionally large models where the costs associated with full model training or even fine-tuning can be prohibitive. Given its relevance, we narrow our discussion to this category. Firstly, quantization refers to the process of reducing the model's footprint by decreasing the bit precision of its weights (Frantar et al., 2022; Yao et al., 2022; Xiao et al., 2022). Quantization not only shrinks the model's size but also accelerates inference, as operations over lower-precision weights are computationally less demanding. Secondly, sparsification, often referred to as pruning, revolves around the concept of selectively removing certain weights elements or masking activation values (Frantar & Alistarh, 2023; Hubara

et al., 2021a;b). The objective behind this is to trim the less salient portions of the model, thereby reducing computational overhead or enhancing model throughput.

Utilizing GPTQ and SparseGPT for model compression, our evaluations, reflected in Figure 2, indicate a decrease in performance with reduced bit counts or parameters, save for the int8 quantization. This contrasts with Frantar et al. (2022)'s claim of unaltered accuracy. On closer inspection, their study heavily leans on perplexity as a primary metric. While they observed some perplexity differences between baseline and quantized models, these differences were minimal compared to their benchmark method Yao et al. (2022). This might explain their understated emphasis on performance on real downstream tasks, highlighting the limitations of relying on perplexity as a model's performance measure in real-world scenarios. Our chosen tasks are selected to evaluate a model's proficiency in fact retrieval, logical understanding, and English language comprehension (elaborated further in Section 3.3). Unlike perplexity, which simply computes the average cross-entropy across the vocabulary, our benchmarks offer a more insightful and interpretable metric. This not only gauges a model's capabilities but also pinpoints deficiencies in specific domains of knowledge.

**Parameter-efficient fine-tuning** There are two prominent strategies for efficient LLM adaptations: adding adapter layers or optimizing the input layer activation. The first strategy revolves around introducing adapter layers, as evidenced by works like Houlsby et al. (2019), Rebuffi et al. (2017), and Pfeiffer et al. (2020). A notable recent contribution, Hu et al. (2021) introduced LoRA – a low-rank adapter that functions as a residual linear path within the feed-forward network. The second strategy, epitomized by studies such as Li & Liang (2021) and Lester et al. (2021), act by refining soft input tokens or prompt embeddings. Among the myriad Parameter-Efficient Fine-Tuning (PEFT) methodologies, our analysis will be primarily on recent innovations like LoRA (Hu et al., 2021), along with prompt-tuning (Lester et al., 2021) and prefix-tuning (Li & Liang, 2021).

## 3 INFERENCE-TIME DYNAMIC PROMPTING

### 3.1 FAILURE MODES OF COMPRESSED LLMs

This study examines failure modes of prompt-tuning in compressed LLM settings. While adhering to the prompt-tuning procedure detailed in Xu et al. (2023), our exploration reveals certain inherent constraints when using perplexity as a performance metric. This is illuminated by Figure 3, which illustrates the average accuracy performance spanning nine tasks juxtaposed with the perplexity (with varied sequence lengths on the 3-bit quantized Llama-7b model). A similar trend, demonstrated in Figure 2, emerges when assessing the reliability of perplexity as a performance metric for compressed models (discussed in Section 2). Interestingly, for shorter prompts, we do observe that for small prompts, there is a generalization between perplexity and task accuracy; this aligns with Xu et al. (2023)'s assumption. However, for longer sequences, an improved perplexity score does not invariably translate to heightened performance. This divergence elucidates the potential pitfalls in prompting at a larger scale.

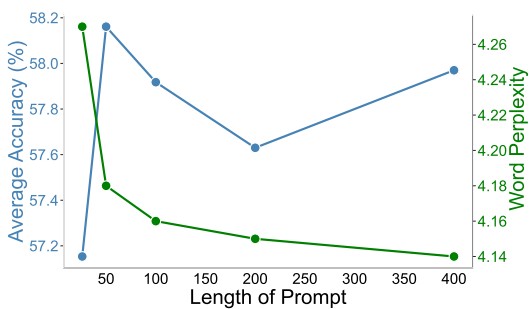

Figure 3: Using a 3-bit quantized Llama-7b model fine-tuned on C4 dataset, we contrast the average accuracy across nine tasks against its word's perplexity score across various prompt lengths. A longer sequence length improves perplexity but does not always sustain better performance.

We posit that the underlying reason for the limited success of prompts is the varying flexibility demands that different inputs necessitate; longer prompts, with their innate rigidity, may struggle to accommodate such versatility. These findings naturally prompt the inquiry into strategies for a more efficient prompting scale. One approach proposed by Lester et al. (2021) suggests prompt tuning can be harnessed as a mechanism for model ensembling. By training $N$ prompts on an identical task, they managed to spawn $N$ distinct models for a task while preserving shared modeling parameters. Their ensemble strategy necessitated input replication across the batch dimension, with each copy

appending to a distinct prompt. Subsequent decision-making employed a majority-rule approach to finalize the output. However, such a technique cannot be scaled efficiently.

Hence, we develop *inference-time dynamic prompting*, a strategy to allow LLMs to still infer with one prompt at a time, but chosen instance-wise from a larger prompt pool.

## 3.2 IDP METHODOLOGY

In prompt tuning, we introduce an additional token sequence, termed as $P$, preceding the input sequence to improve the predicted output likelihood, $Pr_\theta(Y|[P;X])$, where $\theta$ are the static parameters. The sequence $P = p_1, p_2, ...p_n$ is defined by its learnable parameters, $\theta_p \in \mathbb{R}^{n \times e}$, with $n$ being the prompt tokens count and $e$ as their embedding size.

When we extend to a collection of $m$ prompts, represented as $Z = P_1, P_2, ..., P_m$, each prompt has distinct trained parameters. Thus, the modified likelihood of $Y$ becomes $Pr(Y|[Z;X])$. Let's consider the layer-wise token attention as $A \in \mathbb{R}^{b \times h \times tk \times tk}$, where $tk$ stands for the combined token count of $Z$ and $X$. For simplicity, we'll take $b$ and $h$ as one.

To facilitate inference-time dynamic prompting, we introduce two modifications to $A$: **Firstly**, we prevent interactions among the prompts in $Z$ by setting their inter-attention, $A_{[Z_i:Z_j]}$, to $-\infty$. This constraint is twofold: Individual prompts have distinct training and do not share contextual relevance. Mixing them during inference can alter their inherent definitions, affecting the performance. Additionally, by eliminating inter-prompt attention, we can pre-cache the KV (Key, Value) for the prompts, offering dynamic prompt combination capabilities. **Secondly**, for dynamic prompt selection, we measure the mean attention from input-to-prompt and select the prompt attracting the maximum overall input attention: $\arg\max(\{\overline{A}_{[Z_i:X]}|\forall i \in [1, m])$. In the final phase of the self-attention mechanism, we use an attention mask to discard any unintended prompts, ensuring they do not modify the main input sequence. The entire process is depicted in Figure 4.

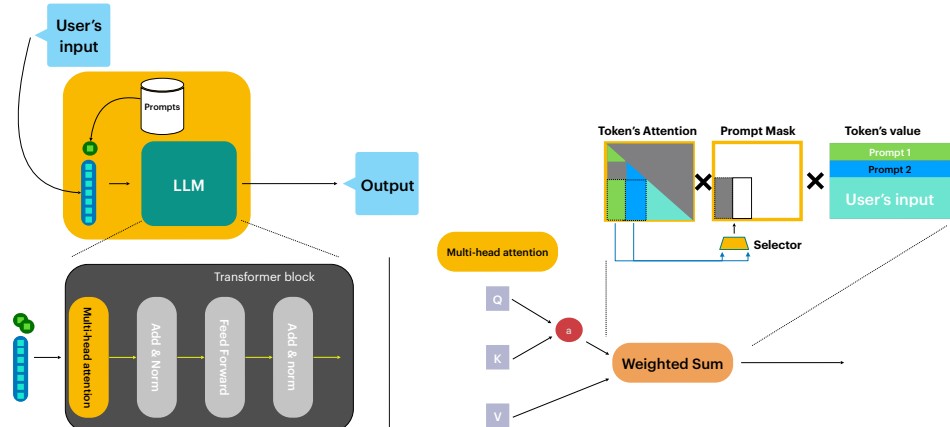

Figure 4: This figure underscores the key advantage of inference-time dynamic prompting (IDP): its minimalistic yet effective design. By making straightforward alterations to the existing weighted sum operation and using the existing attention matrix for prompt selection, IDP accomplishes its objectives without incurring any additional parameter costs.

## 3.3 EXPERIMENTS

**Setup** To facilitate our experimental framework, we employ OPT-6.7b (Zhang et al., 2022) and Llama-7b (Touvron et al., 2023) as foundation models. We subsequently use GPTQ (Frantar et al., 2022) for 3-bit quantization and implement SparseGPT (Frantar & Alistarh, 2023) to prune 50% of the total parameters. We chose these two settings because in our empirical evaluation, as illustrated in Figure 2, they are configurations that impair the model enough to expose the benefit of parameter-efficient fine-tuning. From here, we derive two distinct configurations for each compression technique, each optimized on one of the large-scale text datasets: C4 (Raffel et al., 2020) and

Table 1: This table summarizes the results for 3-bit GPTQ across all nine tasks for multiple fine-tuning baselines and our IDP. World, Common, and Language are performance averages across tasks within those knowledge domains. Average is the average performance across all nine tasks.

| Model | Type | Param | arcE | arcC | sciq | webqs | triviaqa | World | piqa | Common | hellaswag | lambada | winogrande | Language | Average |
|---|---|---|---|---|---|---|---|---|---|---|---|---|---|---|---|
| Llama-7b | — | — | 71.46 | 37.71 | 92.60 | 17.96 | 33.02 | 50.55 | 76.01 | 76.01 | 53.11 | 68.58 | 67.48 | 63.06 | 57.55 |
| Llama-7b | lora | 4.4M | 70.08 | 37.12 | 93.50 | 17.67 | 34.11 | 50.50 | 77.04 | 77.04 | 54.47 | 70.48 | 67.40 | 64.12 | 57.99 |
| Llama-7b | lora | 6.7M | 71.09 | 36.69 | 93.00 | 17.47 | 34.73 | 50.60 | 76.44 | 76.44 | 54.55 | 70.23 | 67.09 | 63.96 | 57.92 |
| Llama-7b | lora | 8.9M | 70.62 | 37.12 | 93.30 | 17.86 | 34.86 | 50.75 | 76.77 | 76.77 | 54.27 | 70.33 | 67.40 | 64.00 | 58.06 |
| Llama-7b | prompt | 0.1M | 71.97 | 38.40 | 92.90 | 20.47 | 33.20 | 51.39 | 75.84 | 75.84 | 53.75 | 69.45 | 67.17 | 63.46 | 58.13 |
| Llama-7b | prompt | 0.2M | 71.51 | 38.31 | 92.10 | 21.11 | 34.56 | 51.52 | 75.84 | 75.84 | 53.92 | 69.69 | 68.75 | 64.12 | 58.42 |
| Llama-7b | prompt | 0.4M | 72.01 | 39.16 | 91.80 | 21.60 | 34.43 | 51.80 | 75.95 | 75.95 | 54.33 | 69.49 | 67.01 | 63.61 | 58.42 |
| Llama-7b | ptune | 3.1M | 70.24 | 36.77 | 91.40 | 14.42 | 30.42 | 48.65 | 75.73 | 75.73 | 53.40 | 66.49 | 63.77 | 61.22 | 55.85 |
| Llama-7b | ptune | 6.5M | 69.57 | 34.81 | 91.30 | 15.55 | 30.65 | 48.38 | 75.30 | 75.30 | 52.98 | 64.84 | 63.22 | 60.35 | 55.36 |
| Llama-7b | ptune | 13.1M | 69.32 | 34.73 | 88.70 | 16.14 | 27.84 | 47.35 | 74.59 | 74.59 | 52.01 | 64.35 | 64.17 | 60.18 | 54.65 |
| Llama-7b | IDP | 0.8M | 72.43 | 39.76 | 92.50 | 19.83 | 36.39 | 52.18 | 76.44 | 76.44 | 53.96 | 70.25 | 67.56 | 63.92 | 58.79 |
| OPT-6.7b | — | — | 64.77 | 29.01 | 89.40 | 9.50 | 17.90 | 42.12 | 75.24 | 75.24 | 48.57 | 65.34 | 63.54 | 59.15 | 51.47 |
| OPT-6.7b | lora | 4.7M | 63.55 | 28.75 | 88.50 | 11.42 | 18.84 | 42.21 | 76.22 | 76.22 | 49.14 | 66.16 | 63.59 | 59.59 | 51.78 |
| OPT-6.7b | lora | 7.1M | 64.27 | 29.01 | 89.20 | 11.07 | 18.95 | 42.50 | 75.90 | 75.90 | 48.89 | 66.50 | 64.40 | 59.93 | 52.02 |
| OPT-6.7b | lora | 9.4M | 64.06 | 29.35 | 88.20 | 13.24 | 18.90 | 42.75 | 76.01 | 76.01 | 49.12 | 66.64 | 63.93 | 59.90 | 52.16 |
| OPT-6.7b | prompt | 0.1M | 64.27 | 28.41 | 89.80 | 10.73 | 18.22 | 42.50 | 76.01 | 76.01 | 49.05 | 65.34 | 63.22 | 59.20 | 51.79 |
| OPT-6.7b | prompt | 0.2M | 64.94 | 28.84 | 89.90 | 10.88 | 18.80 | 42.67 | 75.63 | 75.63 | 49.13 | 65.96 | 63.77 | 59.62 | 51.98 |
| OPT-6.7b | prompt | 0.4M | 64.60 | 28.50 | 89.70 | 11.52 | 18.76 | 42.62 | 76.12 | 76.12 | 48.82 | 65.90 | 63.54 | 59.42 | 51.94 |
| OPT-6.7b | ptune | 3.1M | 63.05 | 28.84 | 89.00 | 10.73 | 18.39 | 42.00 | 75.95 | 75.95 | 48.38 | 64.68 | 60.85 | 57.97 | 51.10 |
| OPT-6.7b | ptune | 6.5M | 62.88 | 28.58 | 88.80 | 10.43 | 18.34 | 41.81 | 75.79 | 75.79 | 48.54 | 65.17 | 60.93 | 58.21 | 51.05 |
| OPT-6.7b | ptune | 13.1M | 62.54 | 29.18 | 88.60 | 10.43 | 18.37 | 41.82 | 75.52 | 75.52 | 48.72 | 65.32 | 63.38 | 59.14 | 51.34 |
| OPT-6.7b | IDP | 0.6M | 64.18 | 28.67 | 90.40 | 11.96 | 19.05 | 42.85 | 76.17 | 76.17 | 49.03 | 66.82 | 63.22 | 59.69 | 52.17 |

Table 2: This table summarizes the results for 50% unstructured sprase using SparseGPT across all nine tasks for multiple fine-tuning baselines and our IDP. World, Common, and Language are performance averages across tasks within those knowledge domains. Average is the average performance across all nine tasks.

| Model | Type | Param | arcE | arcC | sciq | webqs | triviaqa | World | piqa | Common | hellaswag | lambada | winogrande | Language | Average |
|---|---|---|---|---|---|---|---|---|---|---|---|---|---|---|---|
| Llama-7b | — | — | 70.33 | 37.03 | 93.50 | 14.07 | 28.88 | 48.76 | 77.04 | 77.04 | 51.68 | 74.54 | 68.03 | 64.75 | 57.23 |
| Llama-7b | lora | 4.4M | 71.04 | 37.63 | 91.90 | 14.47 | 33.28 | 49.66 | 76.99 | 76.99 | 53.98 | 70.95 | 67.17 | 64.03 | 57.49 |
| Llama-7b | lora | 6.7M | 70.79 | 36.69 | 92.40 | 15.85 | 33.02 | 49.75 | 76.71 | 76.71 | 53.91 | 71.03 | 68.03 | 64.32 | 57.60 |
| Llama-7b | lora | 8.9M | 71.04 | 37.88 | 92.10 | 14.86 | 32.85 | 49.75 | 77.20 | 77.20 | 54.01 | 70.70 | 68.03 | 64.25 | 57.63 |
| Llama-7b | prompt | 0.1M | 71.59 | 38.74 | 93.10 | 15.21 | 29.66 | 49.66 | 77.04 | 77.04 | 53.48 | 71.24 | 67.48 | 64.07 | 57.50 |
| Llama-7b | prompt | 0.2M | 71.38 | 38.57 | 92.20 | 14.86 | 30.48 | 49.50 | 77.15 | 77.15 | 53.75 | 71.76 | 67.09 | 64.20 | 57.47 |
| Llama-7b | prompt | 0.4M | 71.38 | 38.31 | 92.60 | 14.86 | 30.86 | 49.60 | 77.31 | 77.31 | 53.97 | 70.99 | 67.17 | 64.04 | 57.49 |
| Llama-7b | ptune | 3.1M | 63.17 | 32.59 | 88.20 | 11.81 | 24.60 | 44.07 | 72.63 | 72.63 | 50.18 | 64.97 | 56.91 | 57.35 | 51.67 |
| Llama-7b | ptune | 6.5M | 67.17 | 34.90 | 88.70 | 12.11 | 24.74 | 45.52 | 74.76 | 74.76 | 50.36 | 65.59 | 59.12 | 58.36 | 53.05 |
| Llama-7b | ptune | 13.1M | 65.78 | 31.40 | 87.20 | 11.61 | 21.97 | 43.59 | 74.21 | 74.21 | 49.77 | 63.87 | 59.43 | 57.69 | 51.69 |
| Llama-7b | IDP | 0.6M | 72.05 | 39.08 | 92.90 | 14.91 | 30.35 | 49.86 | 77.09 | 77.09 | 53.90 | 70.35 | 67.17 | 63.81 | 57.53 |
| OPT-6.7b | — | — | 63.01 | 28.41 | 89.40 | 9.69 | 17.79 | 41.66 | 75.19 | 75.19 | 47.67 | 70.56 | 63.93 | 60.72 | 51.74 |
| OPT-6.7b | lora | 4.7M | 64.06 | 29.61 | 88.60 | 10.58 | 18.26 | 42.22 | 75.57 | 75.57 | 48.52 | 66.60 | 64.33 | 59.82 | 51.79 |
| OPT-6.7b | lora | 7.1M | 63.93 | 29.78 | 88.20 | 10.14 | 18.48 | 42.11 | 75.90 | 75.90 | 48.58 | 66.45 | 64.56 | 59.86 | 51.78 |
| OPT-6.7b | lora | 9.4M | 62.84 | 29.86 | 88.30 | 10.33 | 18.79 | 42.02 | 75.41 | 75.41 | 48.76 | 66.49 | 65.19 | 60.15 | 51.77 |
| OPT-6.7b | prompt | 0.1M | 63.09 | 28.58 | 90.70 | 12.30 | 18.75 | 42.68 | 75.14 | 75.14 | 48.40 | 68.78 | 63.69 | 60.29 | 52.16 |
| OPT-6.7b | prompt | 0.2M | 63.68 | 29.44 | 90.60 | 12.40 | 18.36 | 42.90 | 75.24 | 75.24 | 48.58 | 67.86 | 63.22 | 59.89 | 52.15 |
| OPT-6.7b | prompt | 0.4M | 64.06 | 29.27 | 89.60 | 12.80 | 19.12 | 42.97 | 75.19 | 75.19 | 48.49 | 67.49 | 63.61 | 59.86 | 52.18 |
| OPT-6.7b | ptune | 3.1M | 61.03 | 28.50 | 86.90 | 13.09 | 19.46 | 41.80 | 72.74 | 72.74 | 46.44 | 62.08 | 59.67 | 56.06 | 49.99 |
| OPT-6.7b | ptune | 6.5M | 63.01 | 29.86 | 88.00 | 9.40 | 17.10 | 41.47 | 75.08 | 75.08 | 47.84 | 64.89 | 61.80 | 58.18 | 50.78 |
| OPT-6.7b | ptune | 13.1M | 60.94 | 29.10 | 88.60 | 13.53 | 19.95 | 42.42 | 73.39 | 73.39 | 46.93 | 62.68 | 62.19 | 57.27 | 50.81 |
| OPT-6.7b | IDP | 0.6M | 64.06 | 29.27 | 89.60 | 12.80 | 19.12 | 42.97 | 75.19 | 75.19 | 48.49 | 67.49 | 63.61 | 59.86 | 52.18 |

Wikitext (Merity et al., 2016). To maintain a controlled experimental space, our fine-tuning of various baseline techniques is restricted to the identical dataset originally used to calibrate our model compression. Subsequently, we report the best results from either configuration.

**Evaluation Tasks**    To gauge the genuine comprehensive performance of LLMs, we identify a suite of evaluation tasks that encapsulate three fundamental domains of cognition: world knowledge, common reasoning, and language understanding. Among the many available tasks, we distilled our focus to a curated list of nine that we deemed most representative.

For the domain of world knowledge, our chosen evaluative tasks were ARC-challenge & ARC-easy (Clark et al., 2018), SCIQ (Welbl et al., 2017), WebQS (Berant et al., 2013), and TriviaQA (Joshi et al., 2017). Tapping into the breadth of language understanding benchmarks, we centered our attention on Hellaswag (Zellers et al., 2019), Lambada (Paperno et al., 2016), and WinoGrande (Sakaguchi et al., 2019). Lastly, for common reasoning, we identified PIQA (Bisk et al., 2019) as our touchstone.

Notably, all the tasks we adopted are structured in a multiple-choice format. Through meticulous internal evaluations, we discerned that the cumulative results procured from these nine tasks are roughly proportional to the model's total parameter count.

Table 3: This table includes results for our Inference-time Dynamic Prompting strategy. To illustrate its effectiveness, we also include the results of the individual prompts used along with naive soft-prompts concatenation

| Model | arcE | arcC | sciq | webqs | triviaqa | **World** | piqa | **Common** | hellaswag | lambada | winogrande | **Language** | **Average** |
|---|---|---|---|---|---|---|---|---|---|---|---|---|---|
| OPT-6.7b/Small | 64.94 | 28.84 | 89.90 | 10.88 | 18.80 | 42.67 | 75.63 | 75.63 | 49.13 | 65.96 | 63.77 | 59.62 | 51.98 |
| OPT-6.7b/Large | 64.02 | 27.90 | 89.50 | 11.32 | 18.37 | 42.22 | 76.39 | **76.39** | 48.81 | 65.42 | 63.22 | 59.15 | 51.66 |
| OPT-6.7b/Concat | 63.80 | 28.50 | 89.40 | 12.30 | 19.55 | 42.71 | 75.79 | 75.79 | 48.92 | 64.72 | 63.85 | 59.16 | 51.87 |
| OPT-6.7b/IDP | 64.18 | 28.67 | 90.40 | 11.96 | 19.05 | **42.85** | 76.17 | 76.17 | 49.03 | 66.82 | 63.22 | **59.69** | **52.17** |
| Llama-7b/Small | 71.97 | 38.40 | 92.90 | 20.47 | 33.20 | 51.39 | 75.84 | 75.84 | 53.75 | 69.45 | 67.17 | 63.46 | 58.13 |
| Llama-7b/Large | 71.51 | 38.31 | 92.10 | 21.11 | 34.56 | 51.52 | 75.84 | 75.84 | 53.92 | 69.69 | 68.75 | 64.12 | 58.42 |
| Llama-7b/Concat | 71.17 | 37.80 | 92.30 | 16.88 | 33.84 | 50.40 | 74.92 | 74.92 | 53.34 | 67.18 | 66.46 | 62.33 | 57.10 |
| Llama-7b/IDP | 71.63 | 38.65 | 92.60 | 21.60 | 33.84 | **51.66** | 76.01 | **76.01** | 53.97 | 69.67 | 68.98 | **64.21** | **58.55** |

**Baseline methods** We gravitated toward three methodologies that stood out in the literature. Specifically, we earmarked Prompt-tuning (Lester et al., 2021), prefix-tuning (Li & Liang, 2021), and LoRA (Hu et al., 2021) as our representative candidates. For the sake of consistent benchmarks across these techniques, we establish the following criteria: 1) The aggregate count of training tokens is standardized at 40,960,000 tokens. Our decision on the total token count draws inspiration from Xu et al. (2023). 2) In alignment with Frantar et al. (2022), we adopt AdamW as our optimization algorithm. Our chosen learning rate stands at 2e-4 with a weight decay set at 1e-5.

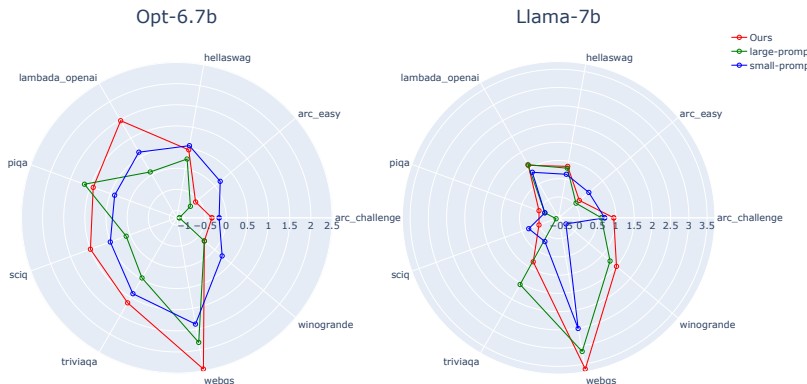

Figure 5: This graph shows the percentage performance improvement using two prompts at various lengths compared to a 3-bit quantized baseline for the OPT and LLama models. We've also show-cased results from our IDP method, which selects prompts dynamically using the same two prompts.

### 3.3.1 IDP RESULTS

For the present study, we used the IDP strategy with two distinct prompts of differing lengths, both trained using the same dataset to streamline our experimental parameters. We subsequently evaluated against our task benchmark, with the comprehensive findings cataloged in Table 3. In a complementary visual aid, Figure 5 highlights the percentage differences in performance against the baseline quantized models, providing an at-a-glance understanding of the performance gains across individual tasks.

Our analysis showed that IDP can bolster average accuracy against other prompting strategies to a notable 0.5% for OPT models and .42% for the Llama models. In contrast, juxtaposing these numbers with the naive prompt concatenation's yield of 0.16% and a decrement of −1.03% underscores the palpable advancement offered by IDP. Elaborating further, Table 3 sheds light on the performance averaged over the different domains of knowledge we previously delineated. Barring the domain of common reasoning for the OPT model, IDP consistently outstripped its counterparts across all knowledge domains.

When pitted against the quantized foundation models, Figure 5 underscores the tasks where IDP manifests superior proficiency. Note that OPT models particularly shine in tasks such as Sciq,

Triviqa, and Webqs, all of which nestle under the world knowledge domain. Conversely, the Llama models evidenced an uptick in tasks like Webqs, Arc, and Winogrand, with gains between 1%-1.5%.

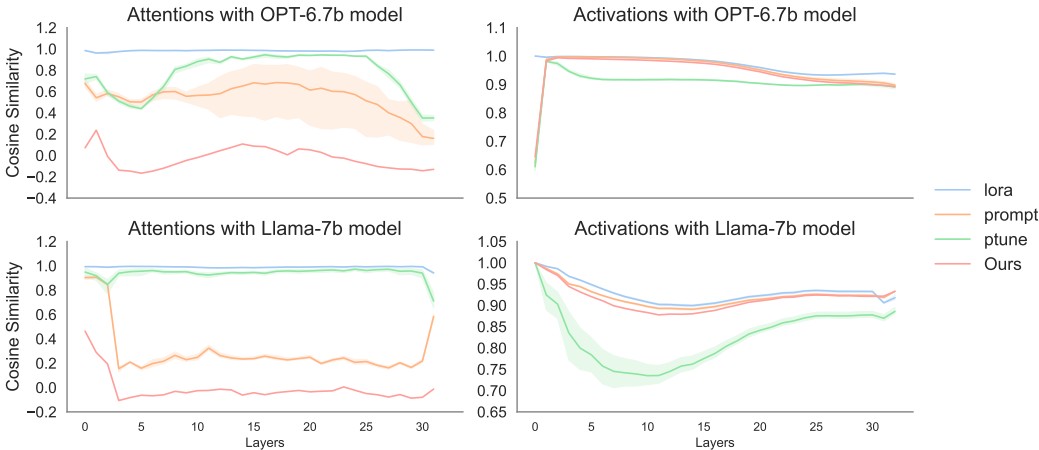

Figure 6: Cosine similarity compares the self-attention and token activation at each layer to an uncompressed baseline using different fine-tuning techniques. A higher cosine score means it's closer to the baseline.

### 3.3.2 MORE COMPARISIONS AND ABLATION STUDIES

We assessed several parameter-efficient fine-tuning baseline methods at diverse size configurations, comparing them with IDP. The findings of our experiments are succinctly captured in Table 1 and Table 2. Based on our analysis, we draw the following insights:

**Performance Recovery:** In the realm of quantization, we observed a clear trend when compared to baseline quantized models: with the exception of prefix-tuning, all other methods—LoRA, prompt, and IDP—uniformly restored performance across our comprehensive set of nine tasks. Notably, the highest average performance increase over the baseline 3-bit models reached 1.24%. Conversely, in the context of sparse models, baseline configurations outperformed all efficient fine-tuning methods in three out of the nine tasks, most prominently in language understanding domains. Nonetheless, the majority of adaptation strategies did achieve above-average task performance, albeit with modest gains over the baseline.

**IDP's Robustness:** We found that IDP consistently outperformed other prompting techniques, demonstrating robust average performance across various settings. Specifically, in quantization scenarios, IDP achieved the highest average performance, equalling or even exceeding its counterparts, including more complex fine-tuning methods like LoRA and prefix-tuning. In additional settings, IDP exhibited performance on par with LoRA, and the set of prompts we used incurred a significantly smaller parameter footprint – being 10x more compact. Further highlighting IDP's capability, Figure 7 underscores its ability to maintain relatively stable high performance at various average prompting sizes.

**Knowledge Domain Adaptation:** Furthermore, results from our study reveal a nuanced

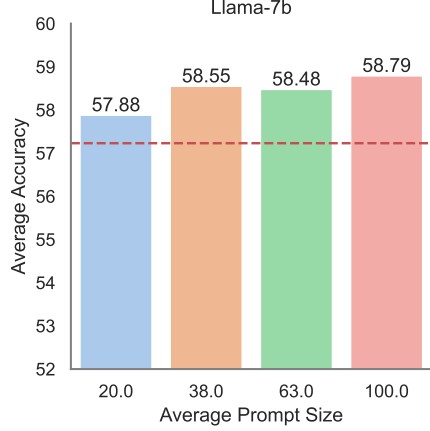

Figure 7: This figure illustrates the average performance over nine tasks using IDP. Results show IDP maintains relatively stable performance working with various average prompt sizes.

improvement landscape. IDP, for instance, exhibits a marked propensity for ameliorating performance on tasks within the domain of world knowledge. However, considering the broader picture, the average-performance distinctions between IDP and the best-performing LoRA model on other domains are **minute**, averaging less than 0.2%. This, coupled with our parameter advantage, makes our performance similar to LoRA's.

## 4    KNOWLEDGE RECOVERING WITH PROMPTING

Earlier, we postulated two conjectures on how prompting recovers performance:

- **H0** (*Null Hypothesis*): Compression culminates in substantial data attrition. In such a scenario, prompt-tuning compensates by incorporating new data sources.

- **H1** (*Alternative Hypothesis*): Compression rechannels, rather than expunges, innate knowledge. This dynamic permits prompt-tuning to spotlight and harness pre-existing acumen.

To test our hypotheses, we visualized the layer-wise attention and activation matrices. We refrained from using magnitude differences and instead used cosine similarity primarily because it allows us to more easily compare differences across layers and provides clearer insights into the true impact of compression on model knowledge. Our elucidations are encapsulated in Figures 6, and 7 from which we deduce the following insights:

- When compared to LoRA, the attention mechanism of both prompting/IDP markedly diverges from the baseline, hinting at a potential contextual redirection. Conversely, the activation patterns echo similarities with LoRA. Given that LoRA incorporates a residual network at every layer to maintain congruity and prompting only at the self-attention, this semblance is unexpected.

- The aforementioned points allude to the capability of prompting/IDP to unearth latent knowledge. Prompting/IDP's ability to unearth extant information is further evidenced by results summarized in Table 1 and Table 2, wherein prompting/IDP manifests a predilection for world knowledge tasks. These tasks, predicated on internalized knowledge repositories within the model, provide further testament to our observation.

- Moreover, IDP exhibits a remarkable consistency in retrieving information. As demonstrated in Figure 7, IDP maintains steady performance across varying average prompt sizes. This consistency suggests that knowledge rerouting can still be effective with fewer tokens, further highlighting potential future optimization and refinements in the deployment of IDP.

- Lastly, our examination of prefix-tuning reveals its inclination to resonate with the original attention patterns. However, its activation patterns deviated significantly, as illustrated in Figure 6, insinuating its inability to redirect knowledge.

Collectively, the evidence heavily suggests H1 as the explanation.

## 5    CONCLUSION

In this study, we delve into prompt tuning as an efficient means of rejuvenating the efficacy of compressed models. Our exploration surfaces certain shortcomings in prior compression-aware prompting endeavors, particularly their inadequacies in effectively catering to complex tasks when scaled up. To address this, we introduce inference-time dynamic prompting (IDP) – a novel approach that facilitates instance-specific prompt selection at inference, all the while abstaining from incurring any additional parameter overheads. From our empirical results, IDP stands out as being significantly more resilient than conventional prompting and LoRA when it comes to recouping performance across diverse knowledge domains. Peeling back the layers on the mechanics of prompting/IDP, we discern that they excel in re-calibrating token attention, effectively harnessing the latent knowledge reservoirs within the models rather than injecting new informational constituents. Our findings argue in favor of the synergy between prompt-tuning and IDP, underscoring their collective capability to restore the performance of models, especially when confronted with complex tasks.

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
