# OpenReview forum: "(Dynamic) Prompting might be all you need to repair Compressed LLMs"
_ICLR.cc/2024/Conference — Submitted to ICLR 2024_

### Official Review · Reviewer_gfXp · 2023-10-16

**Soundness:** 2 fair
**Presentation:** 2 fair
**Contribution:** 2 fair
**Rating:** 3
**Confidence:** 4

**Summary:**

This paper introduces a novel approach to enhance the performance of compressed Language Model Models (LLMs). The proposed method, referred to as Inference-time Dynamic Prompting (IDP), involves adding a set of trainable soft prompts to the user input. During inference, only the prompt with the highest attention score is activated, making it dynamically selected. The authors have empirically validated the efficacy of their approach across nine downstream tasks, demonstrating that IDP effectively restores the performance of two LLMs (LLaMA and OPT) subjected to two compression techniques (quantization and sparsification). The results show that IDP outperforms the baseline models in terms of overall performance.

**Strengths:**

The contribution of this paper is two-fold.

From a theoretical perspective, it delves into the limitations of existing prompting methods by highlighting the deficiency in prompt diversity. Furthermore, it provides an in-depth analysis of the underlying mechanism of IDP, illustrating how IDP effectively "re-channels" the concealed knowledge impacted by compression algorithms.

In the realm of empirical experimentation, IDP demonstrates its superior effectiveness in mitigating the performance degradation caused by compression algorithms when compared to baseline models. Across nine distinct downstream tasks, IDP consistently outperforms these baseline models, which include single prompt, prompt tuning, and LoRA.

**Weaknesses:**

1. Writings

The paper's clarity and organization require improvement. In particular, the flow of the paper can be challenging to follow, particularly in the experiment section. Section 3.3.1 is titled "IDP results," while 3.3.2 is titled "More Comparisons and Ablation Studies," but it is not clear what differentiates these experiments. Additionally, the discussion of Table 1 and Table 2 occurs after the discussion of Table 3, which can be confusing for readers. The suffixes /short and /large in Table 1 and Table 2 are not adequately explained in the main paper. Furthermore, the details of Table 3 and Table 4, specifically the 3 settings of parameter sizes for LoRA, prompt, and ptune, are not provided in either the captions or the main paper.

There are some critical details missing in the paper, particularly concerning hyperparameters. For example, the paper does not specify the prompt length and number, which are denoted as n and m in Section 3.2. Furthermore, there is a lack of information regarding the settings for LoRA, ptuning, and soft prompts. Moreover, it would be helpful if the authors presented the results of the original LLMs to demonstrate the performance drop resulting from compression algorithms.

2. Novelty

The general idea of IDP is based on soft prompting with a dynamic module to select one prompt from a collection of prompts during inference. This solution is akin to the ensemble solution, with the ensembling replaced with a hard selection. Prompt ensembling has been studied by many papers in recent years, and solely applying it to repair compression algorithms isn't novel enough.

3. Performance

The performance enhancement achieved by IDP, in comparison to the baseline models, tends to be relatively modest. Across Table 1 to Table 4, the observed improvement in performance typically falls within the range of 1%. In cases where the performance gap is quite narrow, it may be helpful to conduct a significance test to assess the statistical significance of the results.

4. Hypothesis test

One of the claimed contributions of this paper pertains to the exploration of the reasons behind the performance enhancement achieved by IDP. It introduces hypotheses, denoted as H0 and H1, and conducts hypothesis testing. It's worth noting, however, that this process doesn't adhere to the conventional standard of hypothesis testing, which typically involves the rejection of the null hypothesis (H0). Instead, the authors undertake an analysis of the behavior of IDP and the baseline models, drawing the conclusion that H1 is the more plausible explanation.

The distinction between "data attrition" and "knowledge rechanneling" is somewhat subtle and challenging to precisely define. The evidence presented in the paper, including the shift in attention patterns, may not be compelling enough to definitively support H1, as factors such as the incorporation of new data sources could also lead to changes in attention patterns, making them difficult to reject.

**Questions:**

1. Typo: In page 8, triviqa -> TriviaQA
2. In Table 3, what do /small and /large mean? Do they correspond to "individual prompts"? How long are they?
3. In all experiments, is the performance of the original models comparable? By showing the performance of un-compressed models, we can clearly see how much compression impairs the model performance.

---

> ### Author Response · Authors · 2023-11-18
> **Responding to reviewer gfXp [Part 1 / 2]**
>
> We thank the reviewer for their in-depth response and criticisms. We will try to respond in order of appearance.
>
> **First**, in response to the reviewer’s summary of our work, we wish to clarify that IDP  is an inference process applied to a set of individually fine-tuned prompts. Contrary to what may have been implied in the summary, we do not fine-tune two prompts simultaneously during training to achieve the performance uplift. The uplift reported in Tables 1, 2, and 3 results from directly using two separately fine-tuned prompts. Therefore, IDP can be scaled up with any combination of prompts without training.
>
> **Second**, regarding our manuscript's writing and organization, we will implement the suggested revisions. Notably, sections 3.1.1 and 3.1.2 will be renamed to more clearly delineate their content. In line with your recommendation, we will reorder the Table to match the sequence of discussion. For hyper-parameter clarity, we will update the relevant section to state the numbers explicitly.
> For context, "small" prompts comprise 26 tokens, and "large" prompts comprise 100 tokens. The case of OPT-6.7b and Llama-7b models, the embedding size "e" is 4096. Regarding "m," the number of distinct prompts used, as mentioned in section 3.3.1, is 2, with "n" being 50 and 100. The three prompt sizes employed for the prompting baseline are 25, 50, and 100, consistent across Ptune's three separate settings. LoRA is fine-tuned with hidden dimensions of 2, 3, and 4, and the alpha parameter is set at 36. As suggested, we will also include the baseline uncompressed performance of Llama-7b and OPT-6.7b in table 1 and 2 as well
>
> **Third**, regarding the novelty of our work, we present two perspectives:
> * Similarity to ensembling: While the reviewer is correct to surmise that our method is similar to ensembling, the critical distinction relates to the scalability of IDP. Typically, prompt-ensembling regards each prompt/input as a unique model and would require a number of input replications across the batch dimension and a voting strategy at the end to resolve the different outputs of the same input. This critically impedes scalability. IDP, on the other hand, resolves the selection in a single forward pass.
> * Novelty: To the best of our knowledge, the work most akin to our IDP (Inference Dynamic Prompting) approach is ATTEMPT [1], which similarly employs a combined set of soft prompts. A critical distinction, however, lies in ATTEMPT's methodology. ATTEMPT necessitates additional training specific to targeted tasks to generate a set of weights. These weights are then used to perform weighted averages across the prompt set. This approach involves an extra computational layer and requires further training steps, which IDP efficiently bypasses, thereby streamlining the process. We will include ATTEMPT to our previous work section to showcase our novelty better.
>
> **Fourth**, regarding the significant improvement over baselines, following your suggestion, we have rerun our experimentation using compressed GPTQ Llama-7b for five trials and conducted a T-statistical significant test. The table below marks the T statistical differences between our baselines and IDP and our p-value. Note that since our p-value is below 0.05 and the t-statistic indicates a lower overall sample mean, IDP does produce statistically significant results. In our revision, we will include this ablation study in our result section.
> | model   | type   | size | mean               | std                 | T-stat                 | p-value                |
> |---------|--------|------|--------------------|---------------------|------------------------|------------------------|
> | llama-7b| prompt | 0.1  | 57.454722222222216 | 0.7293593302514931  | -4.132946245610187     | 0.0016425536487376573  |
> | llama-7b| prompt | 0.2  | 57.89944444444445  | 0.7231221120492015  | -1.9204822699592363    | 0.04552389734223814    |
> | llama-7b| prompt | 0.4  | 57.932500000000005 | 0.6070659356954821  | -1.9092908112014755    | 0.04631619380664712    |
> | llama-7b| lora   | 4.4  | 57.89407407407407  | 0.29266610291661366 | -2.602985906153462     | 0.015735718097273196   |
> | llama-7b| lora   | 6.7  | 57.88333333333334  | 0.4033454568766604  | -2.5069461400629938    | 0.01827219202740582    |
> | llama-7b| lora   | 8.9  | 57.91481481481482  | 0.36280647361976315 | -2.36792071959218      | 0.022696752987284078   |
> | llama-7b| IDP    | 0.6  | 58.28222222222222  | 0.6857403424711355  | None                   | None                   |
>
> [1] Peng, Xiangyu, et al. "Model Ensemble Instead of Prompt Fusion: A Sample-Specific Knowledge Transfer Method for Few-Shot Prompt Tuning." The Eleventh International Conference on Learning Representations, 2023, https://openreview.net/forum?id=p0yrSRbN5Bu.

---

> > ### Author Response · Authors · 2023-11-18
> > **Responding to reviewer gfXp [Part 2 / 2]**
> >
> > **Fifth** Regarding the reviewer's criticism of our hypothesis:
> > * Firstly, we acknowledge that our use of the term "hypothesis" might be misconstrued as a reference to a statistical hypothesis. To avoid this confusion, we will revise our introduction and discussion, opting for the term "argument" as a more appropriate descriptor.
> >
> > * Secondly, addressing the distinction between "data attrition" and "knowledge rechanneling," and the potential impact of external data sources on self-attention mechanisms, we reference the example of LoRA, showcased in Figure 6, as a counterpoint. LoRA modifies the weight representation within the self-attention module, a change that should theoretically result in notable alterations in attention patterns. However, our findings reveal that even when LoRA is trained on the same data source as Prompt and Ptune, it exhibits a remarkably high self-attention similarity. Figure 6 presents the average cosine similarity across various size settings as outlined in Tables 1 and 2 for LoRA, Prompt, and Ptune. This consistency in behavior across different sizes suggests that introducing a new data source does not necessarily necessitate a change in attention patterns. Our conclusion is based on the observation that different attention behaviors exist between the two methods. This indicates an alternative approach to recovering performance and output, one that does not rely solely on restoring self-attention patterns.  As a final point, we use Figure 7 to illustrate that reducing the number of tokens can still effectively recover average performance compared to the baseline. This provides further evidence to suggest the possibility of recovering output without necessarily restoring self-attention and that this phenomenon is not strongly related to the number of parameters or data sources.
> >
> > **Question 2**: Long refers to 100 tokens and shorts 50 tokens, and yes, they correspond to the individual prompts.
> >
> > **Question 3**: We will include the performance of the baseline non-compressed model in Tables 1 and 2 for future reference.

---

> ### Comment · Reviewer_gfXp · 2023-11-21
>
> Thanks for the response. I mostly agree with what you pointed out. For the performance part, given the Dynamic Prompt method involves non-trivial additional computation, a marginal improvement, even if statistically significant, is not satisfactory.
>
> Thanks for taking my advise regarding the paper writing. Hope the next version of this paper would be more professional.

---

> > ### Author Response · Authors · 2023-11-22
> > **Following up with reviewer gfXp**
> >
> > As the rebuttal period is nearing its conclusion, we invite the reviewer to respond to our latest reply. We hope we have satisfactorily addressed the reviewer's concerns regarding our work. Should any remaining questions or points of clarification are needed, please do not hesitate to reach out.

---

> ### Author Response · Authors · 2023-11-21
> **Responding to Reviewer gfXp's reply**
>
> We appreciate the reviewer's feedback but we believe there is a **misunderstanding** regarding the cost of operating  IDP.  We emphasize that IDP’s computational costs are **trivial**,  equivalent to a single prompt. In comparison to LoRA, especially in quantized settings, IDP is **2x more efficient** due to its lack of necessity for an additional residual path across linear layers. Regarding parameter size, IDP is **at worst** 14x smaller than LoRA and 21x smaller than Ptune. To further clarify the effectiveness of IDP and clear any confusion, we highlight several key factors that the reviewer may have overlooked:
> * **Trivial Decision Cost** Because we fully take advantage of the existing attention matrix, IDP’s decision process as described in section 3 is trivial as it's simply the row mean.
> * **Efficiency Enhancement** Consider Figure 7, our IDP outperforms a 100-token single prompt of an average accuracy of 58.42 using what is effectively a  single prompt of 38 tokens. This is a **2.6x** improvement in computational efficiency.
> * **KV-Caching Benefits:** If allow for caching, due to the input-invariant nature of prompts, computational costs are effectively **zero**. Because IDP does not allow prompts to intermingle (as described in section 3), there is no need to recompute KV-cache for different combinations of prompts.
>
> In summary, our analysis demonstrates that the operational cost of IDP is trivial. It not only outperforms a naive single prompt approach but also requires fewer tokens. IDP generates performance improvements that are statistically significant when compared to all baselines while costing over one order of magnitude less parameters and saving 2x latency compared to (q)LoRA - **this is the whole picture of “gain”, that we conjecture the reviewer has unintentionally missed to see**. We will ensure that our paper revision explicitly states these points to prevent any future misunderstandings.

---

### Official Review · Reviewer_zxJa · 2023-10-30

**Soundness:** 3 good
**Presentation:** 3 good
**Contribution:** 3 good
**Rating:** 8
**Confidence:** 3

**Summary:**

The work explores the use of dynamic prompting as a tool to improve performance in compressed large language models (LLMs). The authors question the reliability of perplexity as a benchmark for measuring the effectiveness of compressed models and propose dynamic prompting as a solution. They introduce inference-time dynamic prompting (IDP), which selects prompts based on the context of each input. The paper also discusses the displacement of LLM model knowledge during compression and how IDP can utilize this displaced knowledge to enhance performance. Empirical tests demonstrate an average performance improvement across nine tasks.

**Strengths:**

The idea of using inference-time dynamic prompting is interesting, simple and effective.

The paper is clear and easy to follow.

**Weaknesses:**

As the proposed method is quite straightforward, the technical contribution of this paper is not significant.

The idea of using dynamic prompts to enhance downstream task performance is also not novel. The contributions of the method part is limited to a selection mechanism of the dynamic prompts.

**Questions:**

May the author provide further explanation about why the prompt is selected by the maximal mean attention? Why does a large mean attention result in a better prompt?

---

> ### Author Response · Authors · 2023-11-18
> **Responding to reviewer zxJa**
>
> We appreciate the reviewer's support for our paper. In response to their query about why a more significant mean attention might lead to a more suitable prompt for the input, we provide the following explanation. The mean attention metric captures the average token-to-token attention between the input with any given prompt. Our assumption is that prompts eliciting higher levels of activity in response to a given input are likely to be more contextually relevant to that input compared to those that do not. Essentially, what we are measuring is a rough proxy for sequence-to-sequence attention, where higher mean attention values indicate a stronger and potentially more relevant interaction between the prompt and the input sequence.

---

> ### Author Response · Authors · 2023-11-22
> **Following up with Reviewer zxJa**
>
> As the rebuttal period is nearing its conclusion, we invite the reviewer to respond to our latest reply. We hope we have satisfactorily addressed the reviewer's concerns regarding our work. Should any remaining questions or points of clarification are needed, please do not hesitate to reach out.

---

### Official Review · Reviewer_UB9L · 2023-10-31

**Soundness:** 3 good
**Presentation:** 3 good
**Contribution:** 3 good
**Rating:** 6
**Confidence:** 3

**Summary:**

The paper delves into the exploration of prompt tuning as a method to rejuvenate the effectiveness of compressed models. It contrasts this approach with existing methods, especially emphasizing its edge in terms of improving model efficiency without increasing parameters. Through various experiments, it was evidenced that prompt tuning, especially with the proposed IDP (Inference Dynamic Prompting), effectively re-channels the inherent knowledge within models rather than just adding new data. This enables compressed models to perform at par, if not better than their uncompressed counterparts, particularly in world knowledge tasks.

**Strengths:**

Originality: The paper introduces a novel concept called Inference Dynamic Prompting (IDP), standing out as a fresh approach in the realm of compressed models and their performance enhancement.

Quality: The experiments are meticulously carried out, with careful visualization of layer-wise attention and activation matrices. The contrast with other methods like LoRA provides a comprehensive understanding of the efficacy of IDP.

Clarity: The paper is lucidly written, with each section building on the previous, gradually taking the reader from the introduction of the problem to the proposed solution and its validation.

Significance: In a world emphasizing model efficiency, the proposed method is highly relevant. It not only offers a solution to the performance degradation in compressed models but does so without adding extra computational burden, making it significantly impactful in real-world applications.

**Weaknesses:**

While the paper emphasizes the strengths of IDP and prompt tuning, a direct comparison with a broader range of methods, apart from LoRA, would have given a more holistic perspective.

The paper could have benefitted from a more detailed explanation or a separate section dedicated to the underlying theory or intuition behind why IDP works the way it does.

The experiments, though comprehensive, seem to focus predominantly on world knowledge tasks. Incorporating a wider variety of tasks could showcase the versatility of the approach.

**Questions:**

Could the authors elaborate on the foundational theory behind IDP's ability to rechannel inherent knowledge? Is there any theoretical upper limit to its efficiency?

Are there any domains or tasks where IDP might not be as effective, and if so, what are the challenges faced in those scenarios?

Given the effectiveness of IDP in re-channeling latent knowledge within the model, how does it fare in terms of transfer learning across different domains?

---

> ### Author Response · Authors · 2023-11-18
> **Responding to reviewer UB9L**
>
> We thank the reviewer for their support of our paper. As per your advice, we will revise the manuscript to include a section explaining the theory and intuition behind IDP works. But in short,  IDP’s mean attention metric captures the average token-to-token attention between the input with any given prompt. We assume that prompts eliciting higher levels of activity in response to a given input are likely to be more contextually relevant to that input compared to those that do not. Essentially, what we are measuring is a rough proxy for sequence-to-sequence attention, where higher mean attention values indicate a stronger and potentially more relevant interaction between the prompt and the input sequence.
>
> **First** Regarding the foundational theory behind IDP/Prompt’s ability to rechannel inherent knowledge, this is an attribute that we observe and show through deduction.  For instance, we reference the example of LoRA, showcased in Figure 6. LoRA modifies the weight representation within the self-attention module, a change that should theoretically result in notable alterations in attention patterns. However, our empirical findings reveal that even when LoRA is trained on the same data source as Prompt and Ptune, it exhibits a remarkably high self-attention similarity. Figure 6 presents the average cosine similarity across various size settings as outlined in Tables 1 and 2 for LoRA, Prompt, and Ptune. This consistency in behavior across different sizes suggests that introducing a new data source does not necessarily necessitate a change in attention patterns. Our conclusion is based on the observation that different attention behaviors exist between the two methods. This indicates an alternative approach to recovering performance and output, one that does not rely solely on restoring self-attention patterns.
>
> As a final point, we use Figure 7 to illustrate that reducing the number of tokens can still effectively recover average performance compared to the baseline. This provides further evidence to suggest the possibility of recovering output without necessarily restoring self-attention and that this phenomenon is not strongly related to the number of parameters or data sources.
>
> **Second** In addressing whether there are domains where IDP is insufficient, Tables 1 and 2 demonstrate that IDP underperforms in tasks classified under Common Reasoning. These tasks demand the ability to solve problems by comprehending provided information, as opposed to merely recalling information or identifying likely words. The primary challenge here is that the limited number of parameters in the prompts is sufficient only to rechannel attention, thereby recovering information hidden by the compression process. However, these parameters are insufficient to supplement new knowledge that would effectively utilize these facts. This limitation suggests that while IDP can aid in revealing underlying information, its capacity to enhance reasoning or comprehension in more complex task scenarios is constrained.
>
> **Third** For the final question, given that IDP essentially measures the proxy between input and prompt sequences, it is highly applicable that prompts adapt to different tasks. It can be used together in a mix-domain input for increased efficiency. This is something we are actively exploring.

---

> ### Author Response · Authors · 2023-11-22
> **Following up with Reviewer UB9L**
>
> As the rebuttal period is nearing its conclusion, we invite the reviewer to respond to our latest reply. We hope we have satisfactorily addressed the reviewer's concerns regarding our work. Should any remaining questions or points of clarification are needed, please do not hesitate to reach out.

---

### Official Review · Reviewer_dG5d · 2023-11-01

**Soundness:** 2 fair
**Presentation:** 2 fair
**Contribution:** 2 fair
**Rating:** 3
**Confidence:** 3

**Summary:**

This paper focuses on improving compressed LLMs, such as LLMs with 3-bit quantization. The authors argued that the prior work (Xu et al., 2023) suffers from worse performance as the prompt length increases, due to using perplexity as their metric. This paper thus proposed a method called inference-time dynamic prompting (IDP), which learns a collection of prompts and then automatically selects the best prompt during the inference time. Experiments were conducted based on OPT-6.7b and Llama-7b, when 3-bit GPTQ and SparseGPT were applied to compress the LLMs. The results indicate slightly better performance from IDP compared with baselines such as prompt tuning, prefix-tuning, and LoRA.

**Strengths:**

1. The paper studied an important problem of improving compressed LLMs, such that one can strike a better balance between model size/compute and performance.
2. The proposed approach, IDP, was shown to outperform baselines.
3. The discussion about incorporating new knowledge vs. directing knowledge is interesting.

**Weaknesses:**

1. Motivation/intuition: It is not clear to me how the proposed IDP can intuitively address the failure discussed in Section 3.1 (which was said to be caused by the discrepancy between perplexity and accuracy), though I understand that the authors are suggesting to replace long prompts with multiple smaller ones as well as a selection mechanism.
2. For most of the experimental results, IDP outperforms baselines only marginally, and the authors have not conducted any significant tests, which makes the results unconvincing.
3. Lack of clarity and reproducibility: Several experimental or implementation details are missing. For example, what are the sizes of n, e, and m in IDP? What are the prompt sizes for small- and large-prompts in Figure 5?

**Questions:**

Please respond to my concerns in weaknesses.

---

> ### Author Response · Authors · 2023-11-18
> **Responding to reviewer dG5d**
>
> We thank the reviewers for their feedback. We shall address them in the order of appearance.
>
> **First**, we would like to address a slight misunderstanding, as noted by the reviewer in their summary of our work. Contrary to their interpretation, we did not set out to argue that the prior work by Xu et al. experiences diminished performance with increased prompt length. As stated in the third paragraph of our introduction, we found this direction "enticing."  It is because of our interest in their proposed method which led us to discover, as shown in section 3.1, that there is a flaw in their evaluation, which is based solely on perplexity, as we try to scale up our prompt-tuning. Figure 3 shows a region of optimal prompt length in which perplexity correlates to performance. This realization forms the primary motivation for exploring a method that employs a mixture of shorter prompts, as observed from Figure 3, tailored to individual inputs.
>
> With this misconception cleared, we hope this will also address the first question regarding the motivation and intuition of our proposal.
>
> **Second**, regarding the ‘modest’ improvement over baselines, following your suggestion, we have rerun our experimentation using compressed GPTQ Llama-7b for five trials and conducted a T-statistical significant test. The table below marks the T statistical differences between our baselines and IDP and our p-value. Note that since our p-value is below 0.05 and the t-statistic indicates a negative overall sample mean differences, IDP does produce statistically significant results. In our revision, we will include this ablation study in our result section.
> | model   | type   | size | mean               | std                 | T-stat                 | p-value                |
> |---------|--------|------|--------------------|---------------------|------------------------|------------------------|
> | llama-7b| prompt | 0.1  | 57.454722222222216 | 0.7293593302514931  | -4.132946245610187     | 0.0016425536487376573  |
> | llama-7b| prompt | 0.2  | 57.89944444444445  | 0.7231221120492015  | -1.9204822699592363    | 0.04552389734223814    |
> | llama-7b| prompt | 0.4  | 57.932500000000005 | 0.6070659356954821  | -1.9092908112014755    | 0.04631619380664712    |
> | llama-7b| lora   | 4.4  | 57.89407407407407  | 0.29266610291661366 | -2.602985906153462     | 0.015735718097273196   |
> | llama-7b| lora   | 6.7  | 57.88333333333334  | 0.4033454568766604  | -2.5069461400629938    | 0.01827219202740582    |
> | llama-7b| lora   | 8.9  | 57.91481481481482  | 0.36280647361976315 | -2.36792071959218      | 0.022696752987284078   |
> | llama-7b| IDP    | 0.6  | 58.28222222222222  | 0.6857403424711355  | None                   | None                   |
>
> **Third**, we apologize for the lack of clarity with our hyper-parameters, which we will make more explicit in the manuscript. Regarding the sizes of our prompts, small refers to prompts with 26 tokens, and large refers to prompts with 100 tokens. For OPT-6.7b and Llama-7b, the embedding size “e” is 4096. As for m, the number of distinct prompts we mentioned in section 3.3.1 is 2, and n is 50 and 100. The three prompt sizes we used for the prompting baseline are 26, 50, and 100. This is also true for Ptune's three separate settings.

---

> > ### Comment · Reviewer_dG5d · 2023-11-21
> > **Thanks for your response**
> >
> > Thanks to the authors for the response. However, I'm still not quite clear about the connection between Sec 3.1/Figure 3 and IDP. As the authors responded, "their proposed method which led us to discover, as shown in section 3.1, that there is a flaw in their evaluation, which is based solely on perplexity, as we try to scale up our prompt-tuning". But as I can see from the last paragraph on page 4 ("We posit that the underlying reason..."), the discussion has then sharply switched to why longer prompts work or not, and no more discussions on Acc vs. perplexity after that. Honestly, I found the paper not very easy to follow and that the writing needs improvement. In terms of the results, I'm still leaning negative given that IDP only marginally outperforms others.

---

> > > ### Author Response · Authors · 2023-11-22
> > > **Following up with Reviewer dG5d**
> > >
> > > As the rebuttal period is nearing its conclusion, we invite the reviewer to respond to our latest reply. We hope we have satisfactorily addressed the reviewer's concerns regarding our work. Should any remaining questions or points of clarification are needed, please do not hesitate to reach out.

---

> > > > ### Comment · Reviewer_dG5d · 2023-11-23
> > > > **Thank you for the follow up**
> > > >
> > > > Thanks to the authors for further addressing my questions, and I'm also sorry for the late response. In terms of the benefit of parameter size and latency considering the marginal gain, I would hope that the authors could comment on the comparison in Table 3. Generally speaking, I think this work can benefit from another round of revision to improve its clarity and highlight its contribution, but I'm open to adjusting my score based on the post-rebuttal discussion with other reviewers.

---

> > > > > ### Author Response · Authors · 2023-11-23
> > > > > **Responding to reviewer dG5d**
> > > > >
> > > > > Table 3 effectively showcases the capability of IDP  to merge the performance benefits of both Small (26 tokens) and Large (50 tokens) prompts. This combination results in an overall performance that surpasses either prompt used individually and is more efficient than simply concatenating the two prompts. When KV-caching is enabled, the latency improvements observed across the entire testing dataset are roughly equivalent to using a prompt with an average size of 38 tokens.

---

> ### Author Response · Authors · 2023-11-21
> **Responding to reviewer's dG5d response**
>
> We thank the reviewer for their feedback. The connection between Section 3.1/Figure 3 and IDP is straightforward: we demonstrate that scaling prompt-tuning is more effectively achieved with multiple short prompts and dynamic selection, rather than elongating a single prompt. Figure 3 contrasts Perplexity measures with downstream performance, challenging the assumption that better Perplexity always equates to improved task performance, (as assumed in Xu et al). It shows that while longer prompts may improve Perplexity, this does not consistently lead to better performance in downstream tasks. Section 3 explains this by discussing the limitations of longer prompts, particularly their “rigidity” while highlighting the benefit of customizing each input with appropriate shorter prompts. IDP as a result is an efficient way to achieve this scalability while adding **trivial** compute cost. We will make sure the revised manuscript highlights this point more clearly.
>
> Regarding the “marginal” performance improvement as the reviewer put it, we highlight several key efficiency factors that we keep pointing out shall be taken into the context. Firstly, compared to LoRA, particularly in quantized settings, IDP is 2x more efficient. This efficiency is due to IDP's elimination of the need for an additional residual path across linear layers. Concerning parameter size, IDP is at worst 14x smaller than LoRA and 21x smaller than Ptune. Moreover, IDP demonstrates enhanced efficiency. For instance, as shown in Figure 7, our IDP method outperforms a 100-token single prompt, which has an average accuracy of 58.42, using what is effectively a single prompt of 38 tokens. This represents a 2.6x improvement in computational efficiency, emphasizing the **very significant gains** IDP offers on efficiency dimension while being better or comparable on task performance dimension.
>
> In summary,  as we indicated in the last reply, the reviewer was confused on the impression that our performance gain is “trivial” over other methods but seems to overlook the right context that **such performance gain is achieved with 14 times less parameter size and 2 times less latency** (compared to the standard LoRA or (q)LoRA). **This together shall be treated as the highly nontrivial accuracy-efficiency gain of our method**. In order to dive into how much the gain comes up, we dive into the above analysis on several potential factors (e.g., choosing which metric, how long prompt), confirming that a mixture of short prompts is the best option compared to one much longer prompt, and further that such mixture can be cleverly implemented with no overhead at all - that makes the logic line of our paper.
>
> We hope our explanation now has completely clarified any confusion and we’re thankful for your very prompt response!

---

### Meta-Review · Area_Chair_KvVt · 2023-12-07

**Metareview:**

The authors propose a strategy for making LLMs more efficient via inference-time dynamic prompting, or IDP. This approach is based on soft prompts, but here the idea is to keep a collection of these around select which to use at inference time (hence, "dynamically").

This paper offers some nice ideas, but as pointed out by reviewers dG5d, zxJa, and gfXp, the technical approach is not particularly novel. This is not in and of itself a critical issue, but the empirical performance realized using this simple approach is only marginally better than existing methods (it appears to offer an average improvement of ~0.5 points in accuracy over the datasets considered). The authors attempted to address this latter point in response by providing p-values under hypothesis testing, which appears to show "significance" (I'm not clear on how the "pairing" was done here, but that's beside the point); nonetheless, "significance" does not change the marginal mean improvement.

**Justification For Why Not Higher Score:**

The main issue is that the paper offers neither a particularly novel method, nor substantial empirical improvements. Ultimately, I'm left unsure of what the reader might take away from this work.

**Justification For Why Not Lower Score:**

N/A

---

### Decision · Program_Chairs · 2024-01-16

Reject